# Learning A Minimax Optimizer: A Pilot Study

**Jiayi Shen[1*], Xiaohan Chen[2*], Howard Heaton[3*],**
**Tianlong Chen[2], Jialin Liu[4] , Wotao Yin[3,4] , Zhangyang Wang[2]**
[1]Texas A&M University, [2]University of Texas at Austin,
[3]University of California, Los Angeles, [4]Alibaba US, Damo Academy
`asjyjya-617@tamu.edu`,`{xiaohan.chen,tianlong.chen,atlaswang}@utexas.edu`,
`{heaton,wotaoyin}@math.ucla.edu`, `jialin.liu@alibaba-inc.com`

## Abstract

Solving continuous minimax optimization is of extensive practical interest, yet notoriously unstable and difficult. This paper introduces the *learning to optimize* (**L2O**) methodology to the minimax problems for the first time and addresses its accompanying unique challenges. We first present *Twin-L2O*, the first dedicated minimax L2O framework consisting of two LSTMs for updating min and max variables separately. The decoupled design is found to facilitate learning, particularly when the min and max variables are highly asymmetric. Empirical experiments on a variety of minimax problems corroborate the effectiveness of Twin-L2O. We then discuss a crucial concern of Twin-L2O, i.e., its inevitably limited generalizability to unseen optimizees. To address this issue, we present two complementary strategies. Our first solution, *Enhanced Twin-L2O*, is empirically applicable for general minimax problems, by improving L2O training via leveraging curriculum learning. Our second alternative, called *Safeguarded Twin-L2O*, is a preliminary theoretical exploration stating that under some strong assumptions, it is possible to theoretically establish the convergence of Twin-L2O. We benchmark our algorithms on several testbed problems and compare against state-of-the-art minimax solvers. The code is available at: `https://github.com/VITA-Group/L2O-Minimax`.

## 1 Introduction

Many popular applications can be formulated into solving continuous minimax optimization, such as generative adversarial networks (GAN) (Goodfellow et al., 2014), distributionally robust learning (Globerson & Roweis, 2006), domain adaptation (Ganin & Lempitsky, 2014), distributed computing (Shamma, 2008; Mateos et al., 2010), privacy protection (Wu et al., 2018; 2020), among many more. This paper studies such problems: we consider a cost function $f : \mathbb{R}^m \times \mathbb{R}^n \to \mathbb{R}$ and the min-max game $\min_x \max_y f(x, y)$. We aim to find the *saddle point* $(x^*, y^*)$ of $f$:

$$f(x^*, y) \leq f(x^*, y^*) \leq f(x, y^*), \quad \forall (x, y) \in \mathcal{X} \times \mathcal{Y}, \tag{1}$$

where $\mathcal{X} \subset \mathbb{R}^m$ and $\mathcal{Y} \subset \mathbb{R}^n$. If $\mathcal{X} = \mathbb{R}^m$ and $\mathcal{Y} = \mathbb{R}^n$, $(x^*, y^*)$ is called a *global* saddle point; if $\mathcal{X} \times \mathcal{Y}$ is a neighborhood near $(x^*, y^*)$, $(x^*, y^*)$ is a *local* saddle point.

The main challenge to solve problem (1) is the unstable dynamics of iterative algorithms. Simplest algorithms such as gradient descent ascent (GDA) can cycle around the saddle point or even diverge (Benaïm & Hirsch, 1999; Mertikopoulos et al., 2018b; Lin et al., 2019). Plenty of works have been developed recently to address this issue (Daskalakis et al., 2018; Daskalakis & Panageas, 2018; Liang & Stokes, 2019; Mertikopoulos et al., 2018a; Gidel et al., 2018; Mokhtari et al., 2019). However, the convergence is still sensitive to the parameters in these algorithms. Even if the cost function is only changed by scaling, those parameters have to be re-tuned to ensure convergence.

A recent trend of *learning to optimize* (**L2O**) parameterizes training algorithms to be learnable from data, such that the meta-learned optimizers can be adapted to a special class of functions and outperform general-purpose optimizers. That is particularly meaningful, when one has to solve a large number of yet similar optimization problems repeatedly and quickly. Specifically, for existing L2O methods that operate in the space of continuous optimization, almost all of them solve some

---

[*]Equal Contribution.

minimization problem (Andrychowicz et al., 2016; Chen et al., 2017; Li & Malik, 2016), leveraging an LSTM or a reinforcement learner to model their optimizer. Different from classic optimization results that often provide worst-case convergence, most L2O methods have little or no convergence guarantees, especially on problem or data instances distinct from what is seen in training, leaving their *generalizability* in practice questionable (Heaton et al., 2020). Motivated by L2O's success in learning efficient minimization solvers from data, this paper seeks to answer: *whether we could accomplish strong minimax L2O solvers* as well; and if yes, *how generalizable they could be?*

As it might look straightforward at first glance, such extension is **highly nontrivial** due to facing several unique challenges. Firstly, while continuous minimization has a magnitude of mature and empirically stable solvers, for general minimax optimization, even state-of-the-art analytical algorithms can exhibit instability or even divergence. To the best of our knowledge, most state-of-the-art convergence analysis of minimax optimization is built on the convex-concave assumption (Gidel et al., 2018; Mokhtari et al., 2019; Ryu et al., 2019), and some recent works relax the assumption to nonconvex-concave (Lin et al., 2019; 2020). Convergence for general minimax problems is still open. That makes a prominent concern on whether a stable minimax L2O is feasible. Secondly, given the two groups of min and max variables simultaneously, it is unclear to what extent their optimization strategies can be modeled and interact within one unified framework – a new question that would never be met in minimization. Thirdly, the noisy and sometimes cyclic dynamics of minimax optimization will provide noisier guidance (e.g., *reward*) to L2O; not to say that, it is not immediately clear how to define the reward: for minimization, the reward is typically defined as the negative cumulative objective values along the history (Li & Malik, 2016). However, for minimax optimization the objective cannot simply be decreased or increased monotonically.

**Contribution:** This paper is a pilot study into minimax L2O. We start by establishing the first dedicated minimax L2O framework, called *Twin-L2O*. It is composed of two LSTMs sharing one objective-based reward, separately responsible for updating min and max variables. By ablations of the design options, we find this decoupled design facilitate meta-learning most, particularly when the min and max updates are highly non-symmetric. We demonstrate the superior convergence of Twin-L2O on several testbed problems, compared against a number of analytical solvers.

On top of that, we further investigate how to enhance the generalizability of the learned minimax solver[1], and discuss two complementary alternatives with experimental validations. The first alternative is an empirical toolkit that is applicable for general minimax L2O. We introduce *curriculum learning* to training L2O models for the first time, by recognizing that not all problem instances are the same difficult to learn to solve. After plugging in that idea, we show that Twin-L2O can be trained to stably solve a magnitude more problem instances (in terms of parameter varying range). The second alternative explores a theoretical mechanism called *safeguarding*, particularly for the important *special case* of convex-concave problems. When solving a testing instance, safeguarding identifies when an L2O failure would occur and provides an analytical fall-back option (Diakonikolas, 2020). That guarantees convergence for convex-concave problems and, in practice, converges faster even when the problem parameters are drawn from a different distribution from training.

## 2 RELATED WORK

### 2.1 MINIMAX OPTIMIZATION

Following (Neumann, 1928), the problem (1) has been studied for decades due to its wide applicability. Simultaneous gradient descent (SimGD) or gradient descent ascent (GDA) (Nedić & Ozdaglar, 2009; Du & Hu, 2019; Jin et al., 2019; Lin et al., 2019) is one of the simplest minimax algorithms, conducting gradient descent over variable $x$ and gradient ascent over variable $y$. However, the dynamics of SimGD or GDA can converge to limit cycles or even diverge (Benaïm & Hirsch, 1999; Mertikopoulos et al., 2018b; Lin et al., 2019). To address this issue, Optimistic gradient descent ascent (OGDA) simply modifies the dynamics of GDA and shows more stable performance (Daskalakis et al., 2018; Daskalakis & Panageas, 2018; Liang & Stokes, 2019; Mertikopoulos et al., 2018a; Gidel et al., 2018; Mokhtari et al., 2019). OGDA attracts more attention because of its empirical success in training GANs. (Ryu et al., 2019) theoretically studies OGDA by analyzing its continuous time dynamic and

---

[1]We differentiate the usages of two terms: **parameters** and **variables**, throughout the paper. For example, $\min_u \max_v \ ax^2 - by^2$, we call $a, b$ parameters and $x, y$ variables. For simplicity, this paper only discusses the L2O generalizability when the testing instances' parameter distribution differs from the training.

proposes Anchored simultaneous gradient descent that shows good performance. Follow-the-Ridge (Wang et al., 2019) also addresses the limit cycling problem by introducing second-order information into the dynamic of GDA. Lately, K-Beam (Hamm & Noh, 2018) stabilizes the convergence of GDA by duplicating variable $y$, yielding strong performance. At each iteration, it performs gradient ascent independently on $K$ copies of $y$ and greedily chooses the copy that leads to a large function value $f$, then it updates $x$ based on the selected copies.

## 2.2 LEARNING TO OPTIMIZE

As a special instance of meta-learning, L2O has been studied in multiple contexts, with continuous optimization being one of its main playgrounds so far. The first L2O framework is introduced in (Andrychowicz et al., 2016), where both the optimizee's gradients and loss function values are formulated as the input features for an RNN optimizer. Due to the enormous number of parameters, a coordinate-wise design of RNN optimizer is adopted, where all optimization coordinates share the same updating strategy. (Li & Malik, 2016) uses the gradient history and objective values as observations and step vectors as actions in their reinforcement learning framework. (Chen et al., 2017) leverages RNN to train a meta-optimizer to optimize black-box functions. Two effective training tricks, random scaling and objective convexifying, are presented in (Lv et al., 2017). Wichrowska et al. (2017) presents an optimizer of multi-level hierarchical RNN architecture augmented with additional architectural features. Li et al. (2020) introduces a Jacobian regularization to L2O and enhances the domain adaptation performance of optimizees. Chen et al. (2020a) proposes several improved training techniques to stabilize L2O training and ameliorate performance. You et al. (2020); Chen et al. (2020b;c) extend the application scope of L2O into various practical problems such as graph neural network training, domain generalization, and noisy label training.

The above works address continuous minimization problems using single optimizer models. One exception, (Cao et al., 2019), extends L2O to solving Bayesian swarm optimization. The author presents a novel architecture where multiple LSTMs jointly learn iterative update formulas for the swarm of particles, coordinated by attention mechanisms. We also notice that two recent efforts (Jiang et al., 2018; Xiong & Hsieh, 2020) introduce L2O to adversarial training, a renowned application of minimax optimization. However, both of them merely utilize L2O to solve the inner minimization of their minimax problems (i.e., generating attacks), while the outer maximization is still solved analytically. Neither of the two directly solves the full minimax optimization.

# 3 METHOD

## 3.1 MAIN FRAMEWORK: TWIN LEARNABLE OPTIMIZERS (TWIN-L2O)

The main L2O framework we proposed is named *Twin-L2O*, where we use two learnable optimizers to alternate between min and max updates. Our design adopts the basic idea of (Andrychowicz et al., 2016) to use Long Short-Term Memory (LSTM) to model learnable *optimizers*, for solving target problems known as *optimizees*. At each step, LSTM outputs the update of the optimizee variables. The LSTM inputs are typically the current zero-order or first-order information of the optimizee (Andrychowicz et al., 2016; Lee & Choi, 2018), plus the historic optimization trajectory information.

In Twin-L2O, two LSTMs separately update $x$ and $y$ and record historical trajectory information of their own variables respectively. Formally, we consider the minimax problem $\min_x \max_y f(x, y)$. We use two LSTM optimizers, LSTM-Min and LSTM-Max, to updates the min variable $x$ and the max variable $y$ respectively. LSTM-Min is parameterized by $\phi^{\min}$ and LSTM-Max is parameterized by $\phi^{\max}$. At each iteration $t$, Twin-L2O updates $x$ and $y$ in turns and yields the following rule:

$$
\begin{aligned}
x_{t+1} =& x_t + \Delta x_t, \text{where } (\Delta x_t, h_{t+1}^{\min}) = \text{LSTM-Min}\Big( [\nabla_x f(x_t, y_t), \nabla_y f(x_t, y_t)], h_t^{\min}, \phi^{\min} \Big), \\
y_{t+1} =& y_t + \Delta y_t, \text{where } (\Delta y_t, h_{t+1}^{\max}) = \text{LSTM-Max}\Big( [\nabla_y f(x_{t+1}, y_t), \nabla_x f(x_{t+1}, y_t)], h_t^{\max}, \phi^{\max} \Big),
\end{aligned}
\tag{2}
$$

where $h_t^{\min}$ and $h_t^{\max}$ are the historical trajectory information of LSTM-Min and LSTM-Max at time step $t$. This formulation is inspired by the SimGD/GDA-style algorithms (Nedić & Ozdaglar, 2009; Du & Hu, 2019; Jin et al., 2019; Lin et al., 2019) that conduct simultaneous/alternative gradient descent over $x$ and ascent over $y$. Figure A4 (**Appendix A1**) conceptually illustrates the framework.

The next question is to design the L2O reward. To train the LSTM optimizers, the loss function is often to penalize some type of cost, accumulated along the optimization trajectory for a horizon of $T$

steps (also known as the unrolling length for LSTM (Sherstinsky, 2018))

$$\mathcal{L}(\phi^{\min}, \phi^{\max}) = \mathbb{E}_f \left[ \sum_{t=1}^{T} w_t R(f, x_t, y_t) \right],$$ (3)

$w_t$ is chosen to be all 1 following the basic setting in (Andrychowicz et al., 2016), that might be tuned for better performance in future work.

As a key design option, $R(f)$ represents the reward to guide the L2O training. In existing L2O methods for continuous minimization (Andrychowicz et al., 2016; Lv et al., 2017), $R(f)$ is usually simply set to $R(f, x_t)) = f(x_t)$ to encourage fast decrease of objective values over time. To extend this existing reward to the minimax scenario, we cannot directly penalize the overall objective function value either way, since the min and max objectives are entangled. Also, different from pure minimization problems, the Twin-L2O updates (2) consist of two alternating steps governed by two different LSTM optimizers: each accounts for its own subproblem goal (min or max updates), but the two also have to collaborate to explore/exploit the minimax landscape. We specifically design the following reward that implicitly addresses the above issue by setting a new reward function:

$$\mathcal{L}(\phi^{\min}, \phi^{\max}) = \mathbb{E}_f \left[ \sum_{t=1}^{T} \{ [f(x_t, y_{t-1}) - f(x_t, y_t)] + [f(x_t, y_{t-1}) - f(x_{t-1}, y_{t-1})] \} \right].$$ (4)

**Analysis of the reward design**  In Eqn. 4, the first and second terms always characterize two consecutive min and max updates. In more details, the value of $f(x_t, y_t) - f(x_t, y_{t-1})$ solely reflects how effectively the $t$-step max update increases the objective $f$, while $f(x_t, y_{t-1}) - f(x_{t-1}, y_{t-1})$ reflects the effectiveness of $t$-step min update in decreasing the objective $f$. Our goal is then to maximize the weighted accumulated sum for $f(x_t, y_t) - f(x_t, y_{t-1})$, while minimizing the weighted accumulated sum for $f(x_t, y_t) - f(x_t, y_{t-1})$, $t = 1, 2, ..., T$. Combining the two sub-goals together (with a sign change to turn max into min) yields our reward. One may also alternatively interpret Eqn. 4 as penalizing the loss change from $f(x_t, y_t)$ along both $x$ and $y$ updating directions, which would encourage yielding stationary points.

For the reward design, we provide a more detailed discussion in **Appendix A6**. Specifically, we provide a comparison between the *objective-based* reward in Eqn. 4, and another possible *gradient-based* reward. The latter was found to be ineffective in solving the problems presented in Section 4.

**Rationale of the framework selection**  Another important design question is *to what extent learning the min and max updates should be (dis)entangled*: on the one hand, the two steps obviously interact with each other as they jointly explore the minimax landscape; on the other hand, min and max steps commonly have asymmetric difficulty levels, that have been leveraged by previous algorithms. For example, (Hamm & Noh, 2018) demonstrates the failure of alternating gradient descent in minimax optimization due to the multiple solution discontinuity of the inner maximization, and addresses that by simultaneously tracking $K$ candidate solutions for the max step, while the outer minimization remains to take one descent step. Besides the joint reward (4), the default Twin-L2O design leverages two independent LSTMs in Eqn. (2), each dedicatedly handling min or max updates. In comparison, we also consider two other more "entangled" ways: (a) fully entangling the two optimizers, i.e. using one LSTM to simultaneously generate min and max outputs; (b) weakly entangling the two optimizers, by using two LSTMs sharing weights, yet allowing either to maintain its own temporal hidden states. Our ablation experiments (see Section 4.1) find that the default decoupled design in Eqn. (2) seems to facilitate the L2O learning most.

## 3.2 Improving Generalizability of Twin-L2O

Despite the empirical success of L2O, it is unfortunately impossible to ensure that any L2O algorithm always converges. Assuming the objective function type to keep unchanged, the testing instances' parameter distribution may differ from the one of training, and L2O can catastrophically fail. For the Twin-L2O, we discuss two remedies to partially fix this issue and boost its generalizability.

We first propose *curriculum L2O training scheme* as a practical L2O training technique such that Twin-L2O can be trained to work on a much wider coverage of problem parameters than its vanilla versions. That would empirically help the generalizability due to broader coverage by training instance, but would still inevitably fail when meeting unseen testing instances. We then present a

preliminary exploration of the safeguard mechanism on minimax under a special case, i.e., solving convex-concave problems. We demonstrate that with such strong assumptions, it is possible to theoretically establish the "perfect" convergence of Twin-L2O on *any unseen optimizee*.

**Curriculum L2O Training**    When it comes to general minimax problems, it is unlikely to exist an ideal theory to fully ensure Twin-L2O convergence *on all instances*. Therefore, we seek empirical L2O success *of as many instances as possible*. Specifically: *can we train Twin-L2O better, so that it can work on instances at a broader parameter range*?

We find a *curriculum learning* (**CL**) strategy (Bengio et al., 2009) particularly useful. CL was first adopted to train neural networks by first focusing the training on an "easy" training subset (often adaptively selected), that is then gradually grown to the full set. It is known to be effective to stabilize training, especially when the training set is highly varied or noisy (Jiang et al., 2017). Since minimax optimization is notoriously unstable no matter via analytical or learned optimizers, we conjecture that the noisy minimax dynamics might challenge Twin-L2O by providing unreliable guidance and impede its training. Considering that our Twin-L2O is modeled using LSTMs, it is natural to think of whether CL can bring additional gains if applied to meta-training. Previously it was also found effective in L2O for minimization problems (Chen et al., 2020a).

Specifically, in one epoch, we will rank all optimizee instances by their cumulative losses (3) from low to high, and only select the top $C$ instances to count into the total reward. In that way, only the instances that exhibit "good training behaviors" (smaller gradients & more likely to get close to stationary points) will be initially used for updating the Twin-L2O. That prevents the learned optimizer being misled by random failures and outliers, which are commonly found in the early epochs of Twin-L2O training. We by default set the percentage $C$ to start from 20%, then growing linearly every epoch until reaching 100% in the later training stage.

Up to our best knowledge, this is the first effort to incorporate CL with L2O training. We can this Twin-L2O trained with CL as **Enhanced Twin-L2O**: note that it is the same model structure, just trained in a different and better way. More details can be found in **Appendix A4** .

**Safeguard Twin-L2O: A Preliminary Theoretical Exploration**    Most L2O methods have

---

**Method 1** Safeguarded-Twin-L2O for Convex-Concave Saddle Point Problems

1: Initialize $\boldsymbol{u}^1 \in \mathbb{R}^n$, $C \in [0, \infty)$, $\alpha \in (0, \infty)$, $\{\lambda_\ell\} = \{1/(\ell + 1)\}$, $k \leftarrow 2$, weights $\{\phi^k\}$
2: **function** SADDLEHALPERN($\varepsilon$)
3:     $\boldsymbol{u}^2 \leftarrow \frac{1}{2}\left(\boldsymbol{u}^1 + J_{\alpha \partial f}(\boldsymbol{u}^1)\right)$.
4:     **while** $\|\boldsymbol{u}^k - J_{\alpha \partial f}(\boldsymbol{u}^k)\| > \varepsilon$ **do**
5:         $\boldsymbol{z}^{k+1} \leftarrow \text{LSTM}(\boldsymbol{u}^k;\ \phi^k)$
6:                $\triangleleft$ *Apply L2O operator*
7:         **if** $E_{k+1}(\boldsymbol{u}^{k+1}) \leq \dfrac{C}{k+1}$
8:                $\triangleleft$ *Verify safeguard condition*
9:            $\boldsymbol{u}^{k+1} \leftarrow \boldsymbol{z}^{k+1}$
10:                $\triangleleft$ *Use L2O update*
11:         **else**
12:            $\boldsymbol{u}^{k+1} \leftarrow \lambda_k \boldsymbol{u}^1 + (1 - \lambda_k) J_{\alpha \partial f}(\boldsymbol{u}^k)$
13:               $\triangleleft$ *Use fallback update*
14:         $k \leftarrow k + 1$
15:     **return** $\boldsymbol{u}^k$
16: **end function**

---

little or no convergence guarantees. Very recently, a safeguarding mechanism has been introduced to L2O for convex minimization problems with gradient and/or proximal oracles (Heaton et al., 2020). Conceptually, a safeguard is anything that identifies when a "bad" L2O update would occur and what "fallback" update to apply in place of that bad L2O update. In this section, we establish a safeguarding theory and algorithm, specifically for *learned convex-concave saddle point algorithms*. Here the safeguard takes the form of an energy inequality (c.f. Line 6 in Method 1).

In this section, we write $\boldsymbol{u} = (\boldsymbol{x}, \boldsymbol{y}) \in \mathbb{R}^m \times \mathbb{R}^n$ and let $\alpha > 0$. We use the *resolvent*, defined by

$$J_{\alpha \partial f}(\boldsymbol{x}, \boldsymbol{y}) = (\text{Id} + \alpha \partial f)^{-1}, \tag{5}$$

where we note $\partial f = (\partial_{\mathbf{x}} f, -\partial_{\mathbf{y}} f)$. For simple $f$ (e.g., quadratic functions), a closed formula exists for $J_{\alpha \partial f}$. Otherwise, one may use an iterative method to approximate this quantity. In addition, define the residual operator

$$F(\boldsymbol{u}) := \frac{1}{2}\left(\boldsymbol{u} - J_{\alpha \partial f}(\boldsymbol{u})\right), \tag{6}$$

and, for each $k \in \mathbb{N}$, the energy $E_k : \mathbb{R}^m \times \mathbb{R}^n \to \mathbb{R}$ by

$$E_k(\boldsymbol{u}) := \|F(\boldsymbol{u})\|^2 - \frac{\lambda_k}{1 - \lambda_k} \langle F(\boldsymbol{u}), \boldsymbol{u}^1 - \boldsymbol{u} \rangle, \tag{7}$$

where $\{\lambda_k\}$ is a sequence of step sizes. The full method is outlined in the **Method 1**, where the L2O update is denoted by $\mathrm{LSTM}(\boldsymbol{u}^k; \phi^k)$ and the fallback method is a Halpern iteration (Halpern, 1967). Our main result for minimax safeguarding theory is formally stated below:

**Theorem 3.1.** *If the sequence $\{\boldsymbol{u}^k\}$ is generated by Algorithm 1, then*

$$\|\boldsymbol{u}^k - J_{\alpha\partial f}(\boldsymbol{u}^k)\| \leq \frac{1}{2}\left(\frac{d_1}{k} + \sqrt{\frac{d_1^2}{k^2} + \frac{4C}{k}}\right), \quad \textit{for all } k \geq 2, \tag{8}$$

*where $d_1 := \min\{\|\boldsymbol{u} - \boldsymbol{u}^1\| : 0 \in \partial f(\boldsymbol{u})\}$ is the distance from the initial iterate $\boldsymbol{u}^1$ to the set of saddle points and $C \geq 0$ is an arbitrary constant. In particular, this implies each limit point of $\{\boldsymbol{u}^k\}$ is a saddle point.*

Our proof draws and integrates two sources of ideas: (1) the safeguarded L2O technique that has recently just been introduced to convex minimization (Heaton et al., 2020); and (2) Halpern iteration (Diakonikolas, 2020) that is adopted for analytical minimax optimization with favorable theoretical properties. The full proof is provided in **Appendix A3**. Note that this work is **not intended** as a theory innovation on (classical) minimax optimization. Instead, our aim is to extend the emerging idea of safeguarded L2O from convex minimization to convex-concave minimax problems of interest, and shows this idea to be helpful for minimax L2O too: see experiments in Section 4.

## 4 EXPERIMENTS

### 4.1 ABLATION STUDY ON THE DESIGN OF TWIN-L2O

We first investigate the design choices for Twin-L2O that we discussed in Section 3.1. We mainly investigate two aspects: *(i)* whether to share the weights in the two LSTM solvers or not; *(ii)* whether to share the hidden states between the two LSTM solvers or not. That leads us to four options, denoted as (with self-explanatory names): Share-LSTM-Share-Hidden, Share-LSTM-Two-Hidden, Two-LSTM-Share-Hidden, and Two-LSTM-Two-Hidden. We use the seasaw problem, formulated as below, as the testbed for our ablation study (note that the ranges of $a, b$ are picked only to make L2O easy to converge, while more will be investigated in Section 4.3):

$$\text{Seesaw:} \quad \min_x \max_y -by\sin(a\pi x), a \sim U[0.9, 1], b \sim U[0.9, 1] \tag{Seesaw}$$

The Seesaw problem is nonconvex-concave, and is considered challenging (Hamm & Noh, 2018) due to its non-differentiability arising from that the solutions of the state equation or the adjoint state equation are not unique (Danskin, 1966). The L2O training routine follows (Andrychowicz et al., 2016): we use 128 optimizee instances for training; each of them has its parameters i.i.d. sampled, and variables $x, y$ randomly initialized by i.i.d. sampling from $U[-0.5, 0.5]$. A validation set of 20 optimizees is used with parameters and variables sampled in the same way; and similarly we generate a hold-out testing set of another 100 instances. For each epoch, an L2O optimizer will update the optimizee parameters for 1000 iterations, with its unrolling length $T = 10$. When the next epoch starts, all $x, y$ as well as LSTM hidden states are reset. We train the L2O solvers for 200 epochs, using Adam with a constant learning rate $10^{-4}$. We pick the model checkpoint at the epoch when its validation performance reaches the peak. Figure 1 compares the convergence results of the four options, evaluated on the same testing set. We measure the $\ell_2$ distances between the solved variables and their corresponding ground-truth solutions (or the closet one, if multiple exist). It is obvious that only the Two-LSTM-Two-Hidden can successfully converge to the correct solution $(x^*, y^*) = (0, 0)$, which is also the equilibrium. Our major observation from the above experiments is that for minimax L2O optimization, especially for asymmetric problems such as Seesaw, it would be a better choice to use decoupled two LSTM solvers and let them take care of their own trajectory information. We will hence stick to this option and use it as our default Twin-L2O.

All experiments in this and following sections are conducted using the GeForce GTX 1080 Ti GPUs.

### 4.2 COMPARISON WITH STATE-OF-THE-ART ANALYTICAL OPTIMIZERS

In this section, we apply Twin-L2O to two more test problems besides Seesaw:

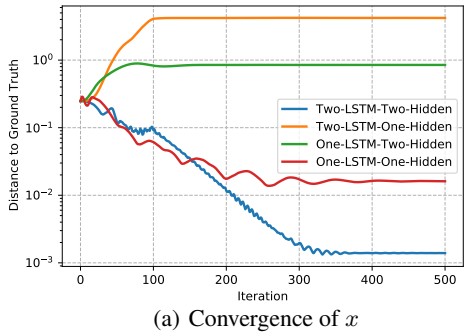
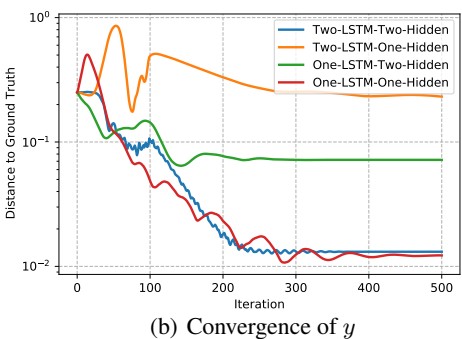

(a) Convergence of $x$          (b) Convergence of $y$

Figure 1: Convergence curves on the ablation study of Twin-L2O design options.

- **Rotated Saddle**[2]: $\min_x \max_y ax^2 - by^2 + 2xy, a \sim U[0.9, 1], b \sim U[0.9, 1]$
- **Matrix Game**: $\min_{\mathbf{x}} \max_{\mathbf{y}} \mathbf{x}^T \mathbf{A} \mathbf{y}, \mathbf{A} \in \mathbb{R}^{5 \times 5}, \mathbf{A}_{i,j} \sim \text{Bernoulli}(0.5) \cdot U[-1, 1]$

On all three problems, we compare Twin-L2O with several state-of-the-art algorithms: Gradient Descent Ascent (GDA) (Lin et al., 2019), Optimistic Mirror Descent (OMD) (Daskalakis et al., 2018) and GD with anchoring (GD-Anchoring) (Ryu et al., 2019). On Rotated Saddle and Seesaw we will compare with $K$-beam (Hamm & Noh, 2018) in addition. For matrix game, we also compare it with the standard Halpern Iteration (Diakonikolas, 2020) that is designed for convex-concave minimax problems. For these analytical methods, all parameters are tuned with careful grid search. We train, validate and test Twin-L2O models following the protocol described in Section 4.1.

Figure 2 plots the convergence curves of all methods, averaged across all testing problems (and each with 20 trials of random $x, y$ initialization). Several observations are drawn below:

- L2O does not show superiority over well-tuned analytical algorithms on the simplest Rotated Saddle problem (and similarly Saddle). The problem is very gradient-friendly, and therefore OMD already achieves the best convergence speed as well as solution quality.

- On Matrix Game, Twin-L2O starts to show competitive edges over analytical solvers with faster convergence speed and higher-precision solutions.

- On the Seesaw problem, Twin-L2O largely outperforms all carefully-tuned analytical algorithms, achieving **one-magnitude higher-precision** solutions with comparable convergence speed. That shows us one *take-home message*: L2O can work for minimax optimization, and can contribute most significantly to those hard problems. That makes minimax L2O a highly meaningful complement to existing analytical minimax solvers. More analysis on comparing the actual computational costs (MAC numbers) can be found in **Appendix A5**.

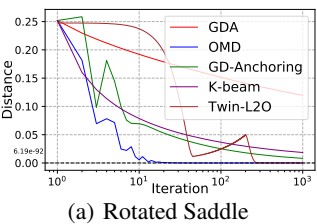
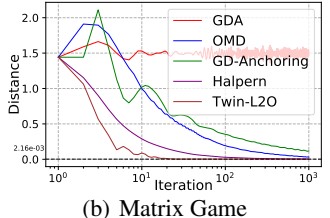
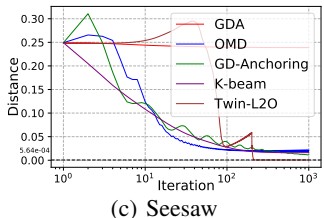

(a) Rotated Saddle       (b) Matrix Game       (c) Seesaw

Figure 2: Convergence comparison of variable $x$ between Twin-L2O and state-of-the-art analytical minimax optimizers (GDA, OMD, GD-Anchoring, and $K$-beam), for three test problems. For results of $y$ variable, please see the **Appendix A2**.

---

[2]We also test on the classical **Saddle** problem, but its behaviors and conclusions are almost identical to the Rotated Saddle. We hence report on Rotated Saddle due to the space limit.

### 4.3 ENHANCED TWIN-L2O: CURRICULUM LEARNING EVALUATION

We again use the Seesaw problem as an example in this section. Its two parameters $a$ and $b$, i.e., the problem period and the scale, are sampled independently from two uniform distributions $U[L_{1a}, L_{2a}]$, $U[L_{1b}, L_{2b}]$. In Section 4.1, both are chosen as $U[0.9, 1]$ for the ease of L2O convergence. We now stretch both parameter ranges and test whether an L2O model can still solve the resultant broader range of problems. All other training protocols follow Section 4.1 identically.

Sections 4.1 and 4.2 evaluate the average solution distances over the testing set (100 instances), which worked fine in the small $[a, b]$ range then. However, when we extend the $[a, b]$ range, we find that the L2O behaviors can differ vastly across testing instances, i.e., some converging quickly while others suffering from heavy fluctuations or even divergence, which is an artifact of inefficient L2O training that leaves it unable to cover the full large problem range. That motivates us to carefully re-design our evaluation metrics here, to reflect both the solution quality and its variation/stability.

For $p$-th testing instance, we record its solution distance $l_2$ $D_t^p$, at epoch $t = 1, 2, ....$ Given two thresholds $\epsilon_{\text{acc}}$ and $\epsilon_{\text{std}}$ (chosen by multi-fold validation; we use default $\epsilon_d = 2 \times 10^{-2}$ and $\epsilon_{\text{std}} = 10^{-4}$), we define two forms of success rate (SR):

$$\text{SR}_1 = \frac{\sum_{p=1}^{n} I(\text{d}(D^p) < \epsilon_{\text{acc}})}{n} \qquad \text{SR}_2 = \frac{\sum_{p=1}^{n} I(\text{Std}(D^p) < \epsilon_{\text{std}})}{n}$$

where $\text{d}(D^p) = \frac{\sum_{t=t_0}^{L} D_t^p}{L - t_0 + 1}$, $\text{Std}(Di) = \text{Std}(\{D_t^p\}_{t=t_0}^{L})$, $t_0 = 0.8L$; $n = 100$ is the number of testing instances; $L = 1000$ is the total iteration number that each instance (optimizee) is trained by L2O. Intuitively, $\text{SR}_1$ emphasizes the average solution precision from the last 20 iterations; and $\text{SR}_2$ measures how large solution variation is seen in the last 20 iterations.

Table 1 compares Twin-L2O and Enhanced Twin-L2O at multiple combinations to stretch the ranges of $a$ and $b$, starting to the original $[0.9, 1] \times [0.9, 1]$, up to as large as $[0, 5] \times [0, 2]$ : the parameter coverage increase by 1,000 times. Adding CL evidently helps Twin-L2O stay effective to train over a broader instance range, under both SR metrics. Vanilla Twin-L2O performs perfectly at $[0.9, 1] \times [0.9, 1]$, yet begins to drop at $[0, 1] \times [0.9, 1]$ (mainly showing higher instability, as indicated by lower $\text{SR}_2$), and hardly succeeds beyond $[0, 3.5] \times [0.9, 1]$. In contrast, Enhanced Twin-L2O obtains nontrivial results even at $[0, 5] \times [0, 1]$ (tens of times wider than the vanilla one).

Table 1: Success rate (SR) of different ranges of $a,b$ on the Seesaw problem

| Settings \ Ranges | a | [0.9, 1.0] | [0.0, 1.0] | [0.0, 3.5] | [0.0, 5.0] | [0.9, 1.0] | [0.0, 5.0] | [0.0, 5.0] |
| | b | [0.9, 1.0] | [0.9, 1.0] | [0.9, 1] | [0.9, 1.0] | [0.0, 1.0] | [0.0, 1.0] | [0.0, 2.0] |
| --- | --- | --- | --- | --- | --- | --- | --- | --- |
| $\text{SR}_1$ (Twin-L2O) | | 100.0% | 96.6% | 0.0% | 0.0% | 97.8% | 11.6% | 22.8% |
| $\text{SR}_2$ (Twin-L2O) | | 100.0% | 89.6% | 17.4% | 16.0% | 96.2% | 31.4% | 71.6% |
| $\text{SR}_1$ (Enhanced Twin-L2O) | | 100.0% | 96.6% | 64.8% | 82.8% | 98.0% | 85.0% | 62.6% |
| $\text{SR}_2$ (Enhanced Twin-L2O) | | 100.0% | 96.0% | 87.8% | 91.2% | 97.4% | 86.2% | 53.8% |

### 4.4 SAFEGUARDED TWIN-L2O EXPERIMENTS

Here we use the matrix game as the example to evaluate the above established safeguard mechanism for convex-concave minimax optimization. We directly take a well-trained Twin-L2O model for matrix game in Section 4.2, where the matrix $\mathbf{A} \in \mathbb{R}^{5 \times 5}$ and $\mathbf{A}_{i,j} \sim \text{Bernoulli}(0.5) \cdot U[-1, 1]$, and the coordinates of initial optimization variables $\mathbf{x}$ and $\mathbf{y}$ are independently sampled from $U[-1, 1]$. During testing, in addition to testing the Twin-L2O model on the testing data from this *seen* distribution, we also evaluate it on *unseen* data, whose $A$ is now sampled from an intentionally very distinct distribution: $\mathbf{A}_{i,j} \sim \text{Bernoulli}(1.0) \cdot U[-8, 8]$. $\mathbf{x}$ and $\mathbf{y}$ are initialized in the same manner.

We compare Safeguarded Twin-L2O (denoted as **Safe-Twin-L2O**) with standard Halpern iteration (Diakonikolas, 2020) as the fallback update, when the L2O update is disapproved in Method 1. We also compare with OMD and GD-Anchoring on both seen and unseen testing data (GDA fails to converge in both cases, even we tune its hyperparameters to our best efforts). The results are shown in Figure 3. When tested on the aggressively varied unseen data, the vanilla Twin-L2O model fails and diverges, but Safe-Twin-L2O remains to converge successfully: even faster than Halpern iteration and OMD, and much better than GD-Anchoring.

## 5 CONCLUSION

This paper studies L2O for minimax optimization for the first time. We present the Twin-L2O model, and further improve its generalizability by introducing a theoretically grounded safeguarding framework (for convex-concave problems), as well as an empirical curriculum training strategy (for general problems). Extensive simulations endorse the promise of our algorithms. This pilot study suggests and paves the way for extending L2O beyond continuous minimization problems.

**Limitation:** The entire L2O field faces challenges to scale up to larger-scale optimization (Andrychowicz et al., 2016), and our study has not yet made an exception. Despite very promising gains from challenging cases such as the Seasaw and Matrix Game problems, the current work only proves the first concept of minimax L2O, on relatively basic and low-dimensional test problems. Our immediate next step is to scale up Twin-L2O, and to explore its potential in solving the minimax application problems of practical interest, such as adversarial training (Jiang et al., 2018; Xiong & Hsieh, 2020) and GANs (Gulrajani et al., 2017). A potential idea might leverage the memory-efficient hierarchical RNN structure in (Wichrowska et al., 2017).

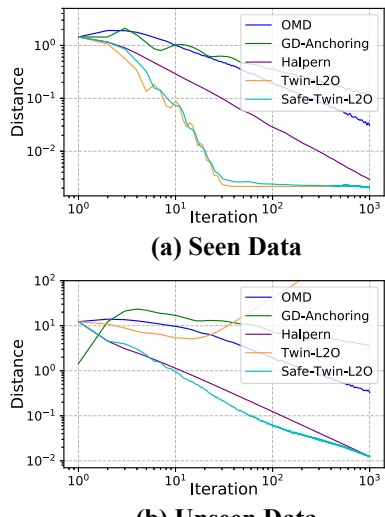

(a) **Seen Data**

(b) **Unseen Data**

Figure 3: Evaluation of Safe-Twin-L2O.

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

## A1 Twin-L2O Framework

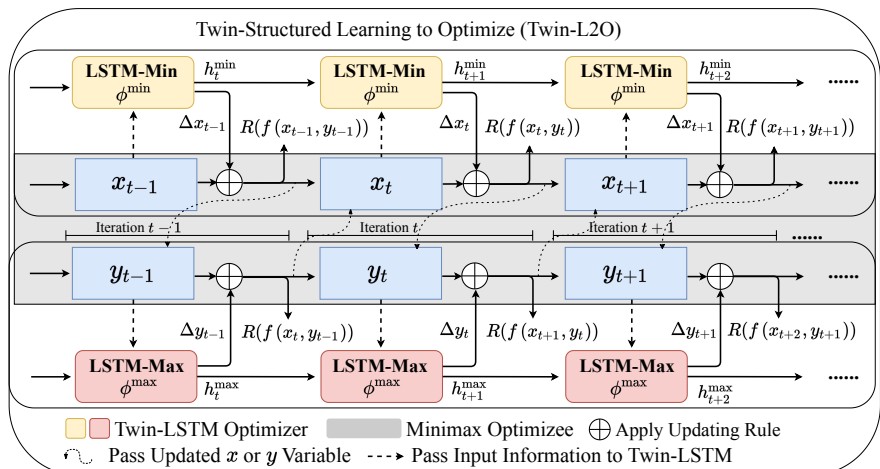

Figure A4: Architecture of Twin-L2O. We let LSTM-Min and LSTM-Max, parameterized by $\phi^{min}$ and $\phi^{max}$, update $x$ and $y$ respectively. As shown by curved dashed lines, Twin-LSTM keeps being updated about the latest variable values of $x$ and $y$ when computing input information and the reward. When constructing the computational graph and training the Twin-LSTM, the solid lines allow gradients to flow while the dashed lines do not pass any gradient (Andrychowicz et al., 2016).

## A2 Comparison with State-of-the-Art Analytical Optimizers

Figure A5 shows the performance of $y$ variable in Rotated Saddle, Matrix Game and Seesaw. The analyses of the results generally align with those in the paper.

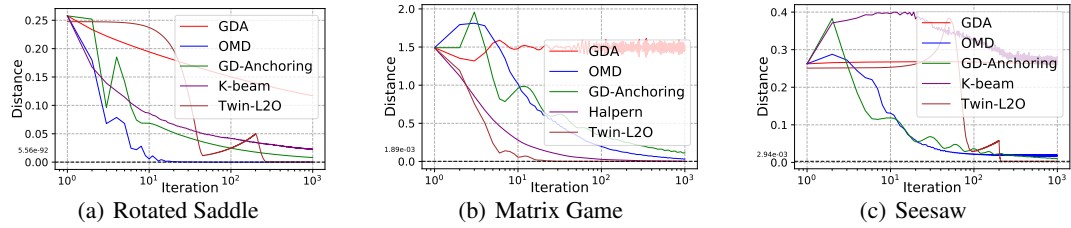

Figure A5: Convergence comparison of variable $y$ between Twin-L2O and state-of-the-art analytical minimax optimizers (GDA, OMD, GD-Anchoring, and $K$-beam), for three test problems.

## A3 Proof of Safeguarding Result

Below is a proof of the main result, Theorem 4.1:

*Proof.* We proceed in the following manner, with *much* credit due to the analysis in (Diakonikolas, 2020). First we verify an inequality with the energy sequence $\{E_k(\mathbf{u}^k)\}$ (Step 1). This is used to obtain the convergence rate (Step 2). Resulting implications about limit points are established last (Step 3).

**Step 1.** We claim

$$E_k(\mathbf{u}^k) \leq \frac{C}{k}, \quad \text{for all } k \geq 2. \tag{9}$$

We proceed by induction. First note $J_{\alpha\partial f}$ is firmly nonexpansive, and so $2F = \mathrm{Id} - J_{\alpha\partial f}$ is also firmly nonexpansive (Bauschke et al., 2011), which implies

$$\|2F(\mathbf{u}) - 2F(\mathbf{v})\|^2 \leq \langle 2F(\mathbf{u}) - 2F(\mathbf{v}), \mathbf{u} - \mathbf{v} \rangle, \quad \text{for all } \mathbf{u}, \mathbf{v} \in \mathbb{R}^m \times \mathbb{R}^n. \tag{10}$$

Using (10) with $\mathbf{u} = \mathbf{u}^2$ and $\mathbf{v} = \mathbf{u}^1$ together with our choice of step sizes $\{\lambda_k\}$, we find

$$E_2(\mathbf{u}^2) = \|F(\mathbf{u}^2)\|^2 - \frac{\lambda_1}{1 - \lambda_1} \langle F(\mathbf{u}^2), \mathbf{u}^1 - \mathbf{u}^2 \rangle \tag{11}$$

$$= \|F(\mathbf{u}^2)\|^2 - \langle F(\mathbf{u}^2), \mathbf{u}^1 - \mathbf{u}^2 \rangle \tag{12}$$

$$= \langle F(\mathbf{u}^2), F(\mathbf{u}^2) - F(\mathbf{u}^1) \rangle \tag{13}$$

$$= \|F(\mathbf{u}^2) - F(\mathbf{u}^1)\|^2 + \langle F(\mathbf{u}^1), F(\mathbf{u}^2) - F(\mathbf{u}^1) \rangle \tag{14}$$

$$\leq \frac{1}{2}\|2F(\mathbf{u}^2) - 2F(\mathbf{u}^1)\|^2 + \langle F(\mathbf{u}^1), F(\mathbf{u}^2) - F(\mathbf{u}^1) \rangle \tag{15}$$

$$\leq \langle F(\mathbf{u}^2) - F(\mathbf{u}^1), \mathbf{u}^2 - \mathbf{u}^1 \rangle + \langle F(\mathbf{u}^1), F(\mathbf{u}^2) - F(\mathbf{u}^1) \rangle \tag{16}$$

$$= -\langle F(\mathbf{u}^2) - F(\mathbf{u}^1), F(\mathbf{u}^1) \rangle + \langle F(\mathbf{u}^1), F(\mathbf{u}^2) - F(\mathbf{u}^1) \rangle \tag{17}$$

$$= 0. \tag{18}$$

Thus, $E_2(\mathbf{u}^2) \leq 0 \leq C/2$, and the base case holds. Inductively, suppose (9) holds taking $k = n$ for some $n \geq 2$. If $\mathbf{u}^{n+1} = \mathbf{z}^{n+1}$, then (9) holds, taking $k = n + 1$, by the conditional statement in Line 6 of Method 1. Alternatively, suppose $\mathbf{u}^{n+1} \neq \mathbf{z}^{n+1}$. Applying (10) with $\mathbf{u} = \mathbf{u}^{n+1}$ and $\mathbf{v} = \mathbf{u}^n$ yields

$$\|F(\mathbf{u}^{n+1}) - F(\mathbf{u}^n)\|^2 \leq 2\|F(\mathbf{u}^{n+1}) - F(\mathbf{u}^n)\|^2 \leq \langle F(\mathbf{u}^{n+1}) - F(\mathbf{u}^n), \mathbf{u}^{n+1} - \mathbf{u}^n \rangle. \tag{19}$$

Upon expansion of the left hand side, we discover

$$\|F(\mathbf{u}^{n+1})\|^2 \leq \langle F(\mathbf{u}^{n+1}), \mathbf{u}^{n+1} - \mathbf{u}^n + 2F(\mathbf{u}^n) \rangle - \langle F(\mathbf{u}^n), \mathbf{u}^{n+1} - \mathbf{u}^n + F(\mathbf{u}^n) \rangle. \tag{20}$$

Algebraic manipulations of the update formula for $\mathbf{u}^{n+1}$ yield the relations

$$\mathbf{u}^{n+1} - \mathbf{u}^n + 2F(\mathbf{u}^n) = \frac{\lambda_n}{1 - \lambda_n}(\mathbf{u}^1 - \mathbf{u}^{n+1}), \tag{21a}$$

$$\mathbf{u}^{n+1} - \mathbf{u}^n + F(\mathbf{u}^n) = \lambda_n(\mathbf{u}^1 - \mathbf{u}^n) - (1 - 2\lambda_n)F(\mathbf{u}^n), \tag{21b}$$

Substituting (21) in (20) gives

$$\|F(\mathbf{u}^{n+1})\|^2 \leq \frac{\lambda_n}{1 - \lambda_n} \langle F(\mathbf{u}^{n+1}), \mathbf{u}^1 - \mathbf{u}^{n+1} \rangle \tag{22}$$

$$- \lambda_n \langle F(\mathbf{u}^n), \mathbf{u}^1 - \mathbf{u}^n \rangle + (1 - 2\lambda_n)\|F(\mathbf{u}^n)\|^2. \tag{23}$$

and we collect terms with $F(\mathbf{u}^{n+1})$ on the left hand side to obtain

$$\|F(\mathbf{u}^{n+1})\|^2 - \frac{\lambda_n}{1 - \lambda_n} \langle F(\mathbf{u}^{n+1}), \mathbf{u}^1 - \mathbf{u}^{n+1} \rangle \leq (1 - 2\lambda_n)\|F(\mathbf{u}^n)\|^2 - \lambda_n \langle F(\mathbf{u}^n), \mathbf{u}^1 - \mathbf{u}^n \rangle. \tag{24}$$

Furthermore, by our choice of step size sequence $\{\lambda_n\}$,

$$1 - 2\lambda_n = \frac{n-1}{n+1} \tag{25}$$

and, for $n \geq 2$,

$$\lambda_n = \frac{n-1}{n+1} \cdot \frac{1}{n-1} = \frac{n-1}{n+1} \cdot \frac{\lambda_{n-1}}{1 - \lambda_{n-1}}. \tag{26}$$

Combining (24), (25), and (26) with the definition of $E_n$ in (7) yields

$$E_{n+1}(\mathbf{u}^{n+1}) \leq \frac{n-1}{n+1} \cdot E_n(\mathbf{u}^n). \tag{27}$$

Applying the inductive hypothesis, we deduce

$$E_{n+1}(\mathbf{u}^{n+1}) \leq \frac{n-1}{n+1} \cdot \frac{C}{n} = \frac{n-1}{n} \cdot \frac{C}{n+1} \leq \frac{C}{n+1}, \tag{28}$$

and this inequality closes the induction. Thus, (9) holds by the principle of mathematical induction.

**Step 2.** Let $\mathbf{u}^\star$ be the projection of $\mathbf{u}^1$ onto $\mathrm{Fix}(J_{\alpha\partial f})$ so that

$$\|\mathbf{u}^1 - \mathbf{u}^\star\| = \arg\min\{\|\mathbf{u}^1 - \mathbf{u}\| : \mathbf{u} \in \mathrm{Fix}(J_{\alpha\partial f})\} = d_1. \tag{29}$$

Note this projection is well defined since the set of saddle points is convex. By (9), for $k \geq 2$,

$$\|F(\mathbf{u}^k)\|^2 \leq \frac{\lambda_k}{1 - \lambda_k} \langle F(\mathbf{u}^k), \mathbf{u}^1 - \mathbf{u}^k \rangle + \frac{C}{k} \tag{30}$$

$$= \underbrace{\frac{\lambda_k}{1 - \lambda_k}}_{=1/k} \left( \langle F(\mathbf{u}^k), \mathbf{u}^1 - \mathbf{u}^\star \rangle + \underbrace{\langle F(\mathbf{u}^k) - F(\mathbf{u}^\star), \mathbf{u}^\star - \mathbf{u}^k \rangle}_{=0} \right) + \frac{C}{k} \tag{31}$$

$$= \frac{1}{k} \langle F(\mathbf{u}^k), \mathbf{u}^1 - \mathbf{u}^\star \rangle + \frac{C}{k} \tag{32}$$

$$\leq \frac{1}{k} \|F(\mathbf{u}^k)\| \|\mathbf{u}^1 - \mathbf{u}^\star\| + \frac{C}{k}. \tag{33}$$

where the third line holds since $F(\mathbf{u}^\star) = 0$ and $F$ is monotone. Using the quadratic formula with the fact that $\|F(\mathbf{u}^k)\|^2 \geq 0$, we obtain (8), as desired.

**Step 3.** Let $\tilde{\mathbf{u}}$ be a limit point of $\{\mathbf{u}^k\}$. This implies there exists a subsequence $\{\mathbf{u}^{n_k}\}$ that converges to $\tilde{\mathbf{u}}$. Since $J_{\alpha\partial f}$ is 1-Lipschitz and norms are continuous, it follows that

$$0 \leq \|\tilde{\mathbf{u}} - J_{\alpha\partial f}(\tilde{\mathbf{u}})\| = \lim_{k \to \infty} \|\mathbf{u}^{n_k} - J_{\alpha\partial f}(\mathbf{u}^{n_k})\| \leq \lim_{k \to \infty} \frac{1}{2} \left( \frac{d_1}{n_k} + \sqrt{\frac{d_1^2}{n_k^2} + \frac{4C}{n_k}} \right) = 0. \tag{34}$$

By the squeeze lemma, we deduce $\tilde{\mathbf{u}} \in \mathrm{Fix}(J_{\alpha\partial f})$, i.e., $\tilde{\mathbf{u}}$ is a saddle point of $f$. Because $\tilde{\mathbf{u}}$ was an arbitrarily chosen limit point, each limit point of $\{\mathbf{u}^k\}$ is a saddle point of $f$. $\qquad\square$

## A4 DETAILS ON CURRICULUM LEARNING

In L2O framework, the reward for training the optimizer is defined as:

$$\mathcal{L}(\phi) = \mathbb{E}_f \left[ \sum_{t=1}^T w_t R(f(x_t)) \right] \tag{35}$$

where $f$ is a distribution of functions. The **Enhanced Twin-L2O** using Curriculum Learning(CL) selects a portion of instances that demonstrate "good training behaviors" (smaller gradients & more likely to get close to stationary points) to be counted into the reward, with the portion $C$ increasing linearly from 20% to 100% as the training epoch increases. In our experiments, the detailed scheme of $C$ is:

$$C = min\{20 + \text{epoch\_index}, 100\}\% \tag{36}$$

where epoch_index denotes the index of epoch when training, starting from 0 and ending with 199 in our case. When applying CL, the actual reward becomes

$$\tilde{\mathcal{L}}(\phi) = \mathbb{E}_f \left[ \sum_{t=1}^T w_t \boldsymbol{q(f)} R(f(x_t)) \right] \tag{37}$$

where $q(f) = 1$ if the value $m(f) = \sum_{t=1}^T w_t \|\nabla_y f(x_t, y_t)\|^2$ ranks top $C$ of all sampled functions, and $q(f) = 0$ otherwise.

This process does not change the structure of Twin-L2O, and essentially acts as adding masks to those training instances that demonstrate poor behavior and ignoring them in the actual training phase. Combining this trick with the existing framework, the Twin-L2O can achieve a higher success rate when solving problems with a larger range of parameters.

## A5    COMPUTATIONAL COST ANALYSIS

We analyze the number of the multiplier–accumulator operation (MAC) of Twin-L2O and K-beam (Hamm & Noh, 2018) for a Seasaw problem testing instance with 20 trials of random $x, y$ initialization, each trial lasting for 1000 iterations. As for K-beam, the numbers of MAC are 2.36M (Million), 3.8M, 8.11 M, 15.31M for K = 1, 2, 5, 10 respectively. For Twin-L2O, the total number of MAC is 3.86M.

We use K = 5 in K-beam for experiments in our paper whose number of MAC costs 2.1 times more than that of Twin-L2O, yet its solution quality in terms of both the converging speed and the precision fails to beat it.

## A6    MORE DISCUSSIONS ON THE DESIGN OF TWIN-L2O REWARD

We term the reward function in Eqn. 4 as an *objective-based* reward, since it penalizes the objective change from $f(x_t, y_t)$ along both $x$ and $y$ updating directions. It is naturally inherited and extends the reward functions prevailing in most prior L2O works for minimization (Andrychowicz et al., 2016; Li & Malik, 2016), whose default reward is to minimize a weighted sum of the past function values.

One may also design the following two rewards, which we name as *gradient-based* rewards:

$$\mathcal{L}(\phi^{\min}, \phi^{\max}) = \mathbb{E}_f \left[ \sum_{t=1}^{T} \| \nabla_x f(x_t, y_t) \|^2 + \| \nabla_y f(x_t, y_t) \|^2 \right], \tag{38}$$

$$\mathcal{L}(\phi^{\min}, \phi^{\max}) = \mathbb{E}_f \left[ \sum_{t=1}^{T} \left( \frac{f(x_t, y_{t-1}) - f(x_t, y_t)}{\|y_t - y_{t-1}\|} \right)^2 + \left( \frac{f(x_{t-1}, y_{t-1}) - f(x_t, y_t)}{\|x_t - x_{t-1}\|} \right)^2 \right]. \tag{39}$$

Eqn. 39 is the gradient-based Nikaido-Isoda function introduced by Raghunathan et al. (2019).

For minimax optimization, it is not immediately clear whether the objective-based or the gradient-based might work practically better. Intuitively by definition, the former is likely to lead towards a saddle point (defined in Eqn. 1) and the latter to a stationary point. They do not always coincide in general, e.g, a stationary point might not be a saddle point. But for all specific test problems we studied in Section 4, a stationary point is also a saddle point.

We try several experiments on the challenging seesaw problem as a specific example, to provide a close comparison between the gradient-based in Eqn. 38 and the objective-based reward. We re-do Twin-L2O by only replacing Eqn. 4 with the gradient-based reward, and our observations are: a) the gradient-based reward solves the seesaw problem worse than the objective-based one; b) the minimization variable $x$ diverges on testing problem instances; c) the maximization variable $y$ will converge to a solution of precision magnitude 0.04 (for reference, $y$ converges to have magnitude less than 0.01 when using the objective-based loss). We further identify one possible cause after analyzing the gradient behaviors. Note here the gradient-based reward could be expressed as:

$$||\nabla_x f(x, y)||^2 + ||\nabla_y f(x, y)||^2 = a^2 b^2 \pi^2 y^2 \cos^2(a\pi x) + b^2 \sin^2(a\pi x) \tag{40}$$

Because $a, b \sim U[0.9, 1]$, the first term often dominates during training due to the $\pi^2$ multiplier, unless $y$ is sufficiently close to zero. The imbalance could be a cause of instability. For example, this reward could sometimes penalize $\cos^2(a\pi x)$ to be close to zero, which is the opposite direction of the true solution $\sin^2(a\pi x) = 0$ . Although this is just a very specific problem example, it reveals that the gradient-based loss may sometimes not work well as expected, due to the instability or asymmetry of min/max gradients.

Besides, we have also tried the second gradient-based reward in Eqn. 39, and find it ineffective. It is mainly because the denominator (consecutive variable differences) can become very small and the loss will then explode and break training.

Back to the objective-based reward used in this paper, we have not observed oscillation empirically from all experiments so far. Our hypothesis is that the recurrent structure of the proposed Twin-L2O

framework (shown in Eqn. 2) plays a role here. Although we use two LSTMs for the min and max updates respectively, the LSTM of one variable actually takes in the information of the other LSTM implicitly, because it takes the output of the other as input. When we penalize the objective function value of one LSTM update, all previous min and max updates can (in principle) be taken into account due to the effect of unrolled back-propagation, e.g., the min and max updates each take reference to not only its own, but also the other's higher-order past trajectory information. While this is a tentative explanation, we think more in-depth analysis of why oscillation may or may not happen in L2O could be a really interesting future work.

Another implicit intuition that leads us to prioritizing the use of objective-based over gradient-based is that, in classic minimization, objective change is summable (i.e., having a finite accumulation), but gradient change is not summable in general (unless with properties such as strong convexity). While summability is itself not a guarantee for good training/testing performance, lack of summability means the loss may have an overly large dynamic range.

To summarize, our function-based objective naturally extends previous L2O convention, works better than other alternatives, and observes no oscillation yet. However, we emphasize that there is no intention to claim the current reward in Eqn. 4 is the best choice for minimax L2O - it is one of plausible options. We do concur the gradient-based reward designs in Eqn. 38 and Eqn. 39 are a complicated yet interesting question, especially when considering more complicated minimax problems. Again, as this paper is intended only as the first work and pilot study towards understanding the profound challenges and rich possibilities for minimax L2O, we believe everything discussed and proposed here, including the loss function, has large room of improvement.

