# OpenReview forum: "Learning A Minimax Optimizer: A Pilot Study"
_ICLR.cc/2021/Conference — ICLR 2021 Poster_

### Official Review · AnonReviewer1 · 2020-10-15
**Interesting but the framework and the models should be better justified**

**Rating:** 6
**Confidence:** 4

**Review:**

### Summary

The paper introduces the _learning to optimize_ (L2O) framework into the solution of minimax problems. The base model is composed of two decoupled LSTMs with a shared objective, with the two LSTMs being respectively responsible for the update of the min and max variables. On top of this, the authors further investigate two possible improvements.  One consists in applying curriculum learning to improve the generalization capability of the solver while the other uses safeguarding to guarantee convergence in convex-concave problems.
Numerical experiments are presented to justify the design choices of the base model and demonstrate the potential of minimax L2O.

---
### Pros
The paper is well-organized, easy to follow and provides a clear context for the problem that is studied.
This problem is particularly challenging and the authors manage to obtain some preliminary results.

---
### Score justification

I do not think the paper meets the acceptance criteria mainly due to the following reasons (all together):
1. Lack of clear motivation.
2. Lack of groundbreaking idea.
3. The definition of the loss function is not convincing.
4. The experiments do not provide strong evidence of the utility of the method either.

Although I fully understand this paper is just intended to be a proof of concept study that demonstrates the usefulness of L2O in minimax problems, I believe the authors should justify more the framework and their algorithmic choices (as done for the decoupled design).

---
### In more detail

1. While finding efficient algorithms for solving minimax problems is without doubt of increasing importance in machine learning today, in the paper there seems to be a lack of justification for the use of L2O in minimax problems.
In the literature, the L2O methodology has been mostly applied to relatively well-studied problems, such as sparse coding and function minimization. On the contrary, minimax optimization is much less well understood and there is little consensus on which optimization algorithm should be used when solving a particular minimax problem. In a sense, while meta learning approaches search for a universal solution to different problems, in minimax optimization people have not even agreed on the solution of a single problem.
Therefore, it would be beneficial if the authors could provide concrete examples for which they believe minimax L2O can really help.
2. In terms of the originality of the paper, the proposed models are basically combinations of existing ideas. While minimax L2O poses unprecedented challenges as claimed by the authors, this work does not seem to propose any dedicated solutions to address these challenges.
3. More importantly, I do not find what the authors propose as the loss function of the solver convincing. The definition of this loss function is probably one of the most important things in the framework. Nonetheless, I fail to see why encouraging stepwise progress in the two variables will necessarily lead to a solution of the problem. In my opinion, the objective (4) may lead to an unstable behavior of the generated iterates.
4. To finish, the experiments do not provide a strong motivation for the use of minimax L2O either.
5. A minor point: the proximal point of the safeguarding mechanism is not always computable so even for convex-concave problems safeguard Twin-L2O does not really offer a practical algorithm.

---

> ### Author Response · Authors · 2020-11-21
> **Response to AnonReviewer1 (1/3)**
>
> We thank you for detailed comments and constructive feedback. We have addressed all your questions below and hope they have clarified all confusions you had on our work. We would really appreciate it if you could kindly re-assess our work, hopefully more positively.
>
> ### Q1: Lack of  justification for the use of L2O in minimax problems.
>
> We humbly yet firmly disagree that the use of L2O in minimax problems fails to be justified. We see very clear motivations to study minimax optimization in the context of L2O, as detailed below.
>
> 1. Using L2O for minimax is *natural* given the success of L2O for minimization optimization. As you also agree, L2O has been successful in the field of “well-studied problems, such as sparse coding and function minimization”. It is natural to extend our scope to cover less-studied problems, and to unleash even more potential of L2O on those difficult cases.
> 2. You are correct “minimax optimization is much less well understood and there is little consensus on which optimization algorithm should be used when solving a particular minimax problem.”. However, this is exactly why L2O shall come into play and we shall see how competitive it could show to be - rather than a reason prohibiting the study of L2O here. In fact, even for nonconvex minimization problems, e.g., training deep networks, there exists no “ consensus on which optimization algorithm should be used” for the best performance either. But L2O has been thoroughly studied in this context too and shown empirical promise.
> 3. We  also emphasize that “meta learning approaches search for a universal solution to different problems” that *come from one common distribution or share common structures*. We expect each different problem class (such as seesaw, matrix game, etc.) + target data distribution would require to train its own L2O. Once the problem type or target data change, the retraining of  L2O essentially re-infers a different black-box learning rule. Therefore, we never assume or have to “agree on the universal solution”.
> 4. Our experimental results on two test functions have already proven significant speedups compared to the SOTA analytical optimizers. Taking the seesaw problem for example, it is nonconvex-concave, and is considered as challenging by past minimax optimization papers (Hamm & Noh, 2018) due to its non-differentiability arising from that the solutions of the state equation or the adjoint state equation are not unique (Danskin, 1966). Twin-L2O then largely outperforms all carefully-tuned analytical algorithms by one-magnitude higher-precision solutions with comparable convergence speed. That shows those hard minimax problems have performance of improvement and L2O can significantly contribute to that.

---

> ### Author Response · Authors · 2020-11-21
> **Response to AnonReviewer1 (2/3)**
>
> ### Q2: Not seem to propose any dedicated solutions to address these challenges.
>
> First of all, we would like to re-emphasize that this paper is poised as the first pilot study of minimax L2O, which has not been investigated prior to this paper. Even so, we have made a set of dedicated contributions to address particular challenges, from model design to training algorithm, that make minimax L2O *non-trivial*.
>
> *The instability of minmax L2O*. For general minimax optimization, even state-of-the-art analytical algorithms can exhibit instability or even divergence. Convergence for general minimax problems is still open. That makes a prominent concern on whether a stable minimax L2O is feasible. Twin-L2O framework, the main contribution of this paper, is exactly what we proposed to address this challenge. Quoting AnonReviewer2, this paper “is a non-trivial effort, as minimax problems are much harder and unstable to solve.”
>
> *Unclarity of the interaction of min and max variables*. This challenge is intensively investigated in Section 4.1, the ablation study of the design choice of minimax L2O, where we come to the conclusion that using twin LSTM solvers sharing a common reward is the best design choice for minimax L2O based on the empirical results. This is also not a trivial design and we consider it as a novel contribution that is specifically used to address the interaction challenge.
>
> *Improved training and preliminary theory for minimax L2O*. The “unprecedented challenges” of training L2O for minimax has further motivated our contribution of two novel training strategies:
>
> 1. We introduce curriculum training into training L2O for the first time. Since minimax optimization is notoriously unstable no matter via analytical or learned optimizers, we conjecture that the noisy minimax dynamics might challenge Twin-L2O by providing unreliable guidance and stuck its training;
> 2. We establish a theoretical mechanism called safeguarding, particularly for the important special case of convex-concave problems. While the safeguarding idea just recently emerged in convex minimization (Heaton et al., 2020), extending the theory and algorithm to convex-concave minimax problems requires theoretical re-work (see Appendix C) and sets a new milestone.
> Overall, we have made a set of dedicated contributions in L2O models, training algorithms and preliminary theory specifically for resolving the unique  challenges in minimax L2O. We hope the above explanation has clarified any misperception of our work.

---

> ### Author Response · Authors · 2020-11-21
> **Response to AnonReviewer1 (3/3)**
>
> ### Q3: Why encouraging stepwise progress in the two variables
>
> We apologize for not having provided more detailed explanations on how we proceed from the minimization L2O reward (3) to the minimax L2O reward (4), but emphasize that was indeed a natural and thoughtful choice. We detail our rationale below, and we are happy to include those discussions into the revised paper if the reviewer finds them convincing enough now.
>
> In Twin-L2O, each of the two LSTMs accounts for its own subproblem goal (min and max updates), while they also have to collaborate to explore/exploit the minimax landscape. The reward (3) for minimization is to minimize a weighted sum of the past function value sequence. To extend (3) for solving minimax, we cannot directly penalize the overall objective function value either way, since we have both min and max entangled.
>
> The proposed loss function (4) is actually a natural extension of (3) to minimax optimization. Expanding (4) for t from 1 to T, one could find it is basically the same as minimizing (S_min - S_max), where S_min is the weighted sum of objective values accumulated after every min step and S_max is the weighted sum of objective values accumulated after every max step.
>
> In the paper, we choose the formulation of (4) to express the loss function because it looks clearer in terms of disentangling the min and max subgoals. For example, the difference of $f(x_{t}, y_{t}) − f(x_{t}, y_{t-1})$ solely reflects how effectively the t-step *max* update *increases* the objective $f$, while $f(x_{t}, y_{t-1}) − f(x_{t-1}, y_{t-1})$ reflects the effectiveness of t-step *min* update in *decreasing* the objective $f$. Our goal is then to maximize the weighted accumulated sum for $f(x_{t}, y_{t}) − f(x_{t}, y_{t-1})$, while minimizing the weighted accumulated sum for $f(x_{t}, y_{t}) − f(x_{t}, y_{t-1}),  t  = 1,2, \dots, T$. Combining the two subgoals together (with a sign change to turn max into min) yields precisely our reward (4).
>
> One may also alternatively interpret (4) as penalizing the loss change from f(xt, yt) along both x and y update directions, which would encourage stationary points.
>
> In sum, we see good motivation and intuition behind our reward (4) and find no obvious flaw in it. While any analytical or L2O minimax optimizer may possibly display “an unstable behavior of the generated iterates” due to the problem itself, we did not yet observe such in our current L2O experiments with the reward (4).
>
> We hope the above clarifications look sufficiently convincing to the reviewer now.
>
> ### Q4: The experiments do not provide a strong motivation for the use of minimax L2O
>
> We respectfully disagree that “the experiments do not provide a strong motivation for the use of minimax L2O either.” We strongly consider that Section 4 has shown solid advantages of minimax L2O from at least three different aspects.
>
> Firstly, well-trained L2O solvers solve minimax problems better than analytical algorithms in terms of speed or the solution quality, or both. On the seesaw problem (Fig 2c), L2O can get one-magnitude higher precision - note that the seesaw problem is nonconvex-concave, and is considered as challenging by past minimax optimization papers (Hamm & Noh, 2018). On the matrix game problem (Fig 2b, Fig 3a), Twin-L2O is tens times faster than the fastest analytical algorithm, Halpern iteration algorithm. Specifically, from Fig 3a we see that Twin-L2O achieves better precision with around 20 iterations than 1,000 iterations of Halpern iteration.
>
> Secondly, training L2O solvers for minimax problems is free from hyperparameter tuning, which is essential for some algorithms to solve well and fast. For example, the hyperparameter K in K-beam algorithm, i.e. the number of candidates that the algorithm keeps track of, needs careful tuning for the success of K-beam. We also empirically observe the convergence of the OMD algorithm is sensitive to the selection of the step size.
>
> Thirdly, we propose two effective extensions to Twin-L2O, Enhanced Twin-L2O and Safeguarded Twin-L2O, which are designed to alleviate two main limitations of the vanilla minimax L2O. In Section 4.3, we see that Enhanced Twin-L2O, an extension based on curriculum learning, significantly increases the success rate of evaluations of minimax L2O on seesaw problems with a much broader instance range. In Section 4.4, Safeguarded Twin-L2O is shown to be able to guarantee the convergence of minimax L2O on problem instances from unseen distributions. These two extensions largely improve the usability of minimax L2O in more general and complicated applications.

---

> > ### Comment · AnonReviewer1 · 2020-11-21
> > **On the loss function**
> >
> > Thank you for your reply. Although I am still not totally convinced, I agree that the decoupled design should be regarded as a novel contribution and the experiments do demonstrate some advantage of minimax L2O with respect to analytical methods.
> >
> > Let me just come back to the definition of the loss function since this is indeed the most important point to me. I now better understand the reasoning behind this formulation. However, I do not find it to be very well-founded. For example, could you comment on why this does not cause the model to produce divergent/cycling or oscillating iterates in order to increase or decrease a lot the value of $f$ at each iteration?
> > If the goal is to find the stationary point, it would for example be much more sensible to minimize $\parallel\nabla_xf(x_t,y_t)\parallel^2+\parallel\nabla_yf(x_t,y_t)\parallel^2$ (another candidate could be the gradient-based nikaido-isoda function [1]). If instead of the gradients we only consider the directional derivatives a very crude approximation would be
> > $$\left(\frac{f(x_t,y_{t-1})-f(x_t,y_t)}{\parallel y_t-y_{t-1}\parallel}\right)^2+\left(\frac{f(x_{t-1},y_{t-1})-f(x_t,y_t)}{\parallel x_t-x_{t-1}\parallel}\right)^2$$
> > instead of what is proposed as the loss function currently. This is why the current definition of the loss does not truly convince me.
> >
> > [1] Raghunathan, A., Cherian, A., & Jha, D. Game Theoretic Optimization via Gradient-based Nikaido-Isoda Function. In ICML 2019.

---

> > > ### Author Response · Authors · 2020-11-22
> > > **More discussion on the loss function**
> > >
> > > Thank you for your timely response and active participation in the discussion! We are happy to discuss the choice of loss functions in more detail. To avoid ambiguity, hereinafter we term the loss function (4) that we use in the original paper as the “objective-based” loss function and the first loss function that you suggest as the “gradient-based”.
> > >
> > > First, the main motivation that we originally chose the objective-based loss (4) was that it naturally inherited and extended the loss functions prevailing in most prior L2O works for minimization (Andrychowicz et al. 2016; Li & Malik, 2016, etc.), whose default loss was to minimize a weighted sum of the past function values (see our last response). Another implicit intuition that led us to prioritizing the use of objective-based over gradient-based is that, in classic minimization, objective change is summable (i.e., having a finite accumulation), but gradient change is not summable in general (unless with properties such as strong convexity). While summability is itself not a guarantee for good training/testing performance, lack of summability means the loss may have an overly large dynamic range.
> > >
> > > However, we also agree that for minimax optimization, it is not immediately clear whether objective-based or gradient-based might work practically better. Intuitively by definition, the former is likely to lead towards a saddle point (defined in Eqn 1) and the latter to a stationary point. They do not always coincide in general, e.g, a stationary point might not be a saddle point. But for all specific test functions we studied, a stationary point is also a saddle point.
> > >
> > > We therefore turn to experiments, on the challenging seesaw problem as a specific example, to provide a close comparison between gradient-based and objective-based. We re-do Twin-L2O by only replacing (4) with the gradient-based loss, and our observations are: a) gradient-based loss solves the seesaw problem worse than the objective-based; b) the minimization variable $x$ diverges on testing problem instances; c) the maximization variable $y$ will converge to have magnitude of 0.04 (for reference, $y$ converges to have magnitude close to 0.01 when using the objective-based loss). We further identify one possible cause after analyzing the gradient behaviors. Note here the gradient-based loss function could be expressed as:
> > > $$\|\nabla_xf(x,y)\|^2+\|\nabla_yf(x,y)\|^2=a^2b^2\pi^2y^2\cos^2(a\pi x)+b^2\sin^2(a\pi x).$$
> > >
> > > Because $a,b\sim U[0.9,1]$, the first term often dominates during training due to the $\pi^2$ multiplier, unless $y$ is sufficiently close to zero. The imbalance could be a cause of instability. For example, this loss function could sometimes penalize $\cos^2(a\pi x)$ to be close to zero, which is the opposite direction to the true solution $\sin^2(a\pi x)=0$. Although this is just a very specific problem example, it reveals that the gradient-based loss may sometimes not work well as expected, due to the instability or asymmetry of min/max gradients.
> > >
> > > Besides, we also tried the second “approximation” loss function you suggested but found it ineffective, mainly because the denominator (consecutive variable differences) can become very small and the loss will then explode and break training.
> > >
> > > Back to the objective-based loss function used in this paper, we have not observed oscillation empirically from all experiments so far. Our hypothesis is that the recurrent structure of the proposed Twin-L2O framework (shown in Eqn 2) plays a role here. Although we use two LSTMs for the min and max updates respectively, the LSTM of one variable actually takes in the information of the other LSTM implicitly, because it takes the output of the other as input. When we penalize the objective function value of one LSTM update, all previous min and max updates can (in principle) be taken into account due to the effect of unrolled back-propagation, e.g., the min and max updates each take reference to not only its own, but also the other’s higher-order past trajectory information. While this is a tentative explanation, we think more in-depth analysis of why oscillation may or may not happen in L2O could be a really interesting future work.
> > >
> > > To summarize, our function-based objective naturally extends previous L2O convention, works so far better than other alternatives, and observes no oscillation yet. However, we emphasize that there is no intention to claim the current loss (4) is the best choice for minimax L2O - it is one of plausible options. We do concur the reviewer the loss function design is a complicated yet interesting question, especially when considering more complicated minimax problems. Again, as this paper is intended only as the first work and pilot study towards understanding the profound challenges and rich possibilities for minimax L2O, we believe everything discussed and proposed here, including the loss function, has the room of improvement and worths more future investigations.

---

> > > > ### Comment · AnonReviewer1 · 2020-11-23
> > > > **Thank you for your explanation**
> > > >
> > > > Thank you for your explanation. I believe it will also help the readers understand better why the current loss is chosen and its possible limitation if a part of our discussion and the additional experiments (some sort of ablation study) are included in an updated version. I indeed think it is far from trivial to answer which loss function shall perform better for minimax L2O, and this depends without doubt also on the problem in question.
> > > >
> > > > I do not have any further questions. Thank you for the responses and I raised my score from 5 to 6.

---

### Official Review · AnonReviewer3 · 2020-10-27
**#Official Review 3**

**Rating:** 7
**Confidence:** 4

**Review:**

Classical iterative minimax optimization algorithms display the unstable dynamics. Their convergence is often sensitive to the parameters and needs to be re-tuned for different problems to ensure convergence. Therefore, there is a practical motivation to develop L2O for minimax problems, so that we could meta-learn and adapt optimization rules to a special class of functions.

To extend L2O from minimization to minimax where two groups of variables need be updated, the authors designed and explored a variety of model options. They find that using two LSTMs, with only their reward function shared, to benefit meta-learning most, particularly when the min and max updates are highly non-symmetric. The decoupled design is aligned with the experience in classical optimizers, e.g., the max step often needs for solution tracking. The authors also described both a curriculum training strategy, and a preliminary theory called safeguarding, to make L2O models be able to solve a wider range of problems.

This paper’s contribution mainly lies in the engineering side, i.e., demonstrating meta learning or L2O can handle more complicated tasks/objectives than conventionally solving minimization. It is an interesting empirical study and is also done solidly. I believe this paper could attract interest and generate follow-up ideas from the L2O community.

On the math side, even though the authors tried to motivate their work from the limitation of classical minimax algorithms, I feel its impact may be limited for the optimization field, as it does not reveal many insights on how to design new minimax algorithms or providing better theory guarantees.

Regarding the experiments, the authors demonstrated three simple testbed functions. As an empirical paper, it would definitely become stronger if the authors can prove their concept on some real minimax problems such as GAN or robust/private training.

The paper is in general well-written. With a lot of contents packed, the authors managed to organize and lay out their logic flow smoothly and clearly. I found just some typos: meta-learing -> meta-learning, draws and integrate -> draws and integrate, recently just introduce -> recently just introduced.

---

> ### Author Response · Authors · 2020-11-21
> **Response to AnonReviewer3**
>
> We thank the reviewer for the positive review and constructive feedback.
>
> ### Q1: Impact.
>
> We first thank the reviewer for kindly commenting on our work as “an interesting empirical study and is also done solidly” and one that “could attract interest and generate follow-up ideas from the L2O community”.
>
> The reviewer is correct that this work is more intended for the meta learning/L2O community, although we hope the L2O behaviors and safeguarding be interpreted to inspire the further minimax optimizer design. The main message we hope to convey is “demonstrating meta learning or L2O can handle more complicated tasks/objectives than conventionally solving minimization”, as the reviewer pointed out. We will make it cleared in the revised paper.
>
> ### Q2: Simple testbed functions.
>
> We would like to humbly point out that some testbed functions we use in the paper are not trivially simple for analytical as well as L2O solvers
> * The seesaw problem is nonconvex-concave, and is considered as challenging by past minimax optimization papers (Hamm & Noh, 2018) due to its non-differentiability arising from that the solutions of the state equation or the adjoint state equation are not unique (Danskin, 1966).
> * On both matrix game and seesaw problems, we also find that learning L2O solvers requires careful model design (as our Section 4.1 ablation study) and training techniques (e.g., Section 4.3).
> * Twin-L2O largely outperforms all carefully-tuned analytical algorithms by one-magnitude higher-precision solutions with comparable convergence speed. That shows those hard minimax problems have performance of improvement and L2O can significantly contribute to that.
>
> We thank the reviewer for the credits of the write-up of this paper and the kind reminders of typos. We will fix all the typos in the updated version.

---

> > ### Comment · AnonReviewer3 · 2020-11-23
> > **I'm satisfied with the rebuttal**
> >
> > Thanks for the detailed rebuttal. It addressed my concerns and I increase my score.

---

### Official Review · AnonReviewer4 · 2020-10-27
**comments on #2099**

**Rating:** 7
**Confidence:** 4

**Review:**

This paper’s main contribution is to extend the L2O framework to solving minimax problems for the first time.

Minimax optimization is in general unstable and harder to solve, challenging whether an L2O model can indeed figure out effective learning rules from data. Further, in order to design L2O for minimax problems, one has to decide to what extent the learning models for min and max updates should be coupled, and what reward shall be used (minimizing the negative cumulative objective value is no longer viable)

By discussing and comparing a number of design options, the authors find that two decoupled LSTMs sharing one variation-based reward is the best empirical design. They show this minimax L2O can display favorable empirical convergence speed on several testbed problems, compared against a number of analytical solvers.

More importantly, most L2O methods have little or no convergence guarantees, which constitutes another roadblock for broadening their practical usage, such as people often questioning whether they will diverge on some even slightly different problem or data. The authors presented Safeguarded Twin L2O, a preliminary theory effort saying that under some strong assumptions, it is possible to theoretically establish the general worst-case convergence of Twin-L2O.

The proof draws and integrate two sources of ideas: (1) the safeguarded L2O technique recently developed for convex minimization (Heaton et al., 2020); and (2) Halpern iteration. For this part, it is unclear to me why Halpern iteration was chosen as the fallback method in their safeguarded L2O, since it is not a current popular or fast minimax solver. Is the safeguarded L2O framework also compatible with other convergent minimax solvers?

In Section 4.4, the authors said on unseen data, their safe-Twin-L2O remains to converge successfully and even faster than Halpern iteration. Is this really correct? As far as I understand, on an unseen distribution the optimization should “fall back” to exactly the Halpern iteration; so shouldn’t safeguarded L2O behave identically with Halpern on unseen data?

---

> ### Author Response · Authors · 2020-11-21
> **Response to AnonReviewer4**
>
> We thank the reviewer for the detailed review and positive feedback.
>
> ### Q1: Why was Halpern iteration chosen as the fallback method in their safeguarded L2O?
>
> Halpern's method is a suitable choice for fallback methods for three reasons.
>
> 1. Given a fixed point operator (e.g., a proximal or gradient operator), it is incredibly simple/easy to implement.
>
> 2. The anchoring term yields more stability in iterations so that they oscillate less around fixed points than they might otherwise.
>
> 3. Halpern iterations yield the optimal rate of convergence. In general, if the update operator T is simply averaged (e.g., a gradient descent or proximal operator), then the residual norm |u^k - T(u^k)| converges at a rate O(1/sqrt(k)). However, by using Halpern iteration, the anchoring term improves the result to the optimal rate O(1/k).
>
> ### Q2: The experiment in Section 4.4
>
> We kindly refer the reviewer to the conditional statement in Line 7 of Method 1. If an energy inequality holds, then the Twin-L2O update is executed. Since the Twin-L2O update is trained on the seen distribution, we should not expect it to perform well on an unseen distribution. Consequently, the safeguard is likely (but not always) to be activated. The reason the Safe-Twin-L2O plot in Figure 3b outperforms Halpern is because there are some updates (although likely infrequent) where Twin-L2O updates are performed rather than Halpern updates.

---

> > ### Comment · AnonReviewer4 · 2020-11-23
> > **Thanks for the rebuttal**
> >
> > The rebuttal addresses my concern and I decide to keep my score unchanged.

---

### Official Review · AnonReviewer2 · 2020-10-27
**Reviews for L2O**

**Rating:** 7
**Confidence:** 4

**Review:**

This paper studies the learning to optimize (L2O) for minimax optimization. Since L2O has been studied in a few works, extending L2O from continuous minimization to minimax is a straightforward idea and not super-novel. But it also is a non-trivial effort, as minimax problems are much harder and unstable to solve.

The authors proposed to use two LSTMs with one shared reward, for updating min and max variables respectively. They presented a careful ablation study of design options such as (semi-)weight sharing between the two and their reward function, which is valuable for helping us understand what matters for L2O to work in minimax L2O.

The authors then presented two extensions to improve the generalization of Twin-L2O. The first one is based on curriculum learning to focus the meta-training gradually from easy to hard instances.  The second one is a minimax safeguard mechanism under a special case of solving convex-concave problems; the theory part seems to be a direct extension of (Heaton et al., 2020).

The following suggestions are for the authors:

-	It is impressive to see that on relatively challenging minimax problems such as Seesaw, Twin-L2O can achieve one-magnitude higher-precision solutions than carefully tuned analytical algorithms. The number of iterations and MAC numbers needed for convergence are also comparable. I wonder whether the authors could also make a fair comparison on their running clock time?

-	One further suggestion is that, it would be natural (and to the authors’ good) to combine enhanced L2O and safeguarded L2O together for solving convex-concave problems, so that we can get an impression on how large benefits it can lead to if we combine the best of the two L2O improvement ideas.

-	I appreciate the authors clearly and openly discussed the current work’s limitations by end of the paper. Although the paper was positioned as “proof of concept”, it could also be strengthened if some real problem can be demonstrated, e.g., training of a very simple GAN or so.

---

> ### Author Response · Authors · 2020-11-21
> **Response to AnonReviewer2**
>
> We thank the reviewer for the positive review and constructive feedback.
>
> ### Q1: Comparison of running clock time.
>
> We first thank the reviewer for his/her appreciation of the superior performance of Twin-L2O over carefully tuned analytical algorithms. It is a great suggestion to also compare the running clock time of Twin-L2O with analytical algorithms. The comparison follows the protocol below.
>
> We apply Twin-L2O and K-Beam to 1,000 different problem instances, one by one, i.e. the batch size for Twin-L2O is 1, and the reported running clock time is the average over the 1,000 problems instances. We only record the time elapsed within the computation period, excluding other overheads such as I/O time. Note that we use Pytorch for the implementation of Twin-L2O with GPU acceleration. However, GPU provides little acceleration for K-Beam on 1-D seesaw problems. The experiments are conducted in the same environment on one machine with one NVIDIA Tesla V100 GPU. The comparison of running clock time per problem instance (in seconds) is shown below. We could see a clear linear relation between the running clock time of K-Beam and the selection of $K$. The running clock time of Twin-L2O is at the same scale as K-Beam and is between $K=10$ and $K=15$, which we perceive as a positive result. Bear in mind that the final precision of Twin-L2O achieves one-magnitude higher-precision than K-Beam. We also think Twin-L2O has more potential of acceleration with larger batches, while we only use a single sample per batch. We will include this comparison of running clock time in the updated version.
>
> |       Method       | Twin-L2O | K-Beam K=5 | K-Beam K=10 | K-Beam K=15 | K-Beam K=20 |
> |:------------------:|:--------:|:----------:|:-----------:|:-----------:|:-----------:|
> | Running Clock Time |   0.46   |    0.20    |     0.33    |    0.50    |     0.68    |
>
>
> ### Q2: Combine enhanced L2O and safeguard L2O.
>
> Thank you for this great suggestion. Since the seesaw problem is not convex-concave, we turn to the matrix game problem and significantly enlarge the range of problem instances that the L2O solvers will see during training and apply Enhanced L2O, and then we apply during testing the safeguard mechanism during testing. The results will be included in the final version.

---

> > ### Comment · AnonReviewer2 · 2020-11-23
> > **Thanks for the clarification.**
> >
> > My questions were well addressed. I think this work is solid and I recommend  acceptance.

---

### Decision · Program_Chairs · 2021-01-07
**Final Decision**

**Decision:**

Accept (Poster)

**Comment:**

The paper proposed Twin L2O (learning to optimize) for extending L2O from minimization to minimax problems. The authors honestly discussed the limitation of Twin L2O and proposed two improvements upon it with better generalization/transferability. While some reviewer had some concerns on the motivation of applying L2O to solve minimax problems and the motivation of the loss-function design (why objective-based one is chosen but not gradient-based one), the authors have done a particularly good job in the rebuttal. Even though this is more a proof-of-concept paper, it indeed has novel and solid contributions, and should be accept for publication.